# Molecular Mechanism of Lipotoxicity as an Interesting Aspect in the Development of Pathological States—Current View of Knowledge

**DOI:** 10.3390/cells11050844

**Published:** 2022-03-01

**Authors:** Katarzyna Lipke, Adriana Kubis-Kubiak, Agnieszka Piwowar

**Affiliations:** Department of Toxicology, Faculty of Pharmacy, Wroclaw Medical University, 50-367 Wroclaw, Poland; adriana.kubis-kubiak@umw.edu.pl (A.K.-K.); agnieszka.piwowar@umw.edu.pl (A.P.)

**Keywords:** lipotoxicity, fatty acid, oxidative stress, insulin resistance, inflammation

## Abstract

Free fatty acids (FFAs) play numerous vital roles in the organism, such as contribution to energy generation and reserve, serving as an essential component of the cell membrane, or as ligands for nuclear receptors. However, the disturbance in fatty acid homeostasis, such as inefficient metabolism or intensified release from the site of storage, may result in increased serum FFA levels and eventually result in ectopic fat deposition, which is unfavorable for the organism. The cells are adjusted for the accumulation of FFA to a limited extent and so prolonged exposure to elevated FFA levels results in deleterious effects referred to as lipotoxicity. Lipotoxicity contributes to the development of diseases such as insulin resistance, diabetes, cardiovascular diseases, metabolic syndrome, and inflammation. The nonobvious organs recognized as the main lipotoxic goal of action are the pancreas, liver, skeletal muscles, cardiac muscle, and kidneys. However, lipotoxic effects to a significant extent are not organ-specific but affect fundamental cellular processes occurring in most cells. Therefore, the wider perception of cellular lipotoxic mechanisms and their interrelation may be beneficial for a better understanding of various diseases’ pathogenesis and seeking new pharmacological treatment approaches.

## 1. Introduction

The term “lipotoxicity” was first adopted in 1994 by Lee et al. in the context of elucidating the pathogenesis of obesity-related β-cell alterations both before and at the onset of type 2 diabetes mellitus (T2DM) [1]. This report was the first to associate increased plasma free fatty acids (FFA) with insulin resistance (IR) and β-cell unresponsiveness to hyperglycemia. Currently, due to further research, lipotoxicity is defined as the harmful effect of high concentrations of lipids and lipid derivatives manifested as a set of metabolic disorders in the cells of non-fatty tissues, causing disturbances in their metabolism and/or loss of function or apoptosis. This phenomenon most often affects the cells of the pancreas, liver, skeletal muscles, heart muscle, and kidneys. The cell of non-adipose tissues and organs, such as β-cells, hepatocytes, cardiomyocytes, podocytes, and myocytes, are able to store lipids to a limited extent. The mechanisms involved in lipotoxicity (also referred to as lipotoxic effects) include oxidative stress (OS), the stress of the endoplasmic reticulum (ER), induction of IR, or inflammation. Lipotoxicity is relatively well understood in the organs mentioned above, while there are very few comprehensive descriptions of the effects of lipotoxicity on cell metabolism. Although cell metabolism is based on the same pathways in every cell type, tissue-specific cell differences must be taken into account.

The molecular mechanisms of fatty acids’ action vary due to chemical classification as saturated or unsaturated. The research confirms that most deleterious effects are provided by saturated fatty acids (such as palmitic acid, also referred to as palmitate), whereas unsaturated fatty acids relieve cells from lipotoxic effects. Oleic acid, containing one unsaturated bond in its structure, was shown to be more steatogenic but less damaging than palmitic acid. Furthermore, oleic acid prevents oxidative stress and apoptosis caused by palmitic acid in human hepatocytes and rat and mouse myocytes [2,3,4]. This review focuses on the deleterious effects of saturated fatty acids on the molecular level; therefore, the term ‘FFA’ is understood in this article as saturated free fatty acids. Additionally, when mentioning the duration of FFA exposure, the term ‘acute’ is understood as less than 6 h; however, the term ‘chronic’ is understood as longer than 10 h.

A review of current knowledge about cellular mechanisms of lipotoxicity was performed. Although there are many existing reviews about lipotoxicity concerning the aforementioned organs, this is, to the best of our knowledge, the first review focusing on the description of the cellular mechanism underlying lipotoxicity.

## 2. The Role of Homeostasis Disturbances of Fatty Acids in Lipotoxicity

Fatty acids (FAs) occur in the organism in two main forms—free fatty acids (FFAs) circulating in plasma (bound to albumin as a lipoprotein particle) and triacylglycerol (TAG) depot in cytoplasm inside cells. After infiltrating the cell membrane through various transporters, FFAs are either oxidized in the mitochondria to generate adenosine triphosphate (ATP) or esterified in the cytoplasm with a glycerol particle, thus creating TAG. Research distinguishes three main types of fatty acids transporters: fatty acid translocase (FAT, also known as cluster of differentiation 36, CD36, or scavenger receptor), fatty acid binding protein (FABP), and fatty acid transport protein (FATP). Several subtypes of those transporters are specified depending on the main site of expression, e.g., FATP5 in the mouse liver, FATP2 in the mouse kidney, and human adipocyte FABP (A-FABP) [5,6,7]. The importance of those transporters comes down to regulating fatty acid influx by their increased or decreased expression. There is numerous evidence that lipid accumulation is connected to the expression of fatty acid transporters. Induced transcriptional activation of CD36 is associated with lipid accumulation in mouse hepatocytes, and overexpression of heart-type FABP in mouse podocytes disturbs lipid metabolism and aggravates inflammation and oxidative stress [8,9]. However, the knockout of FATP2 ameliorates mouse kidney fibrosis, and the downregulation of CD36 prevents lipid accumulation in rat cardiomyocytes and mouse macrophages [6,10,11]. FATPs and CD36 have also been recognized as contributing factors to the progression of nonalcoholic fatty liver disease (NAFLD) in a human clinical study [10].

Lipids are accumulated in the cell in the form of lipid droplets—the cellular organelles coated with proteins called perilipins that regulate the rate of exposition of lipids to lipases. There are five members of the perilipin family (perilipin 1–perilipin 5) that are expressed mainly in adipose tissue [11]. However, perilipin 5 is also associated with mitochondria in cells and tissues, such as mouse myocytes or mouse hepatocytes [12,13]. In the state of increased energy demand, TAG can be hydrolyzed back to FFA. The two processes regulating fatty acid homeostasis are lipolysis and lipogenesis, and the intensity of those processes is regulated by insulin, catecholamines, or adipose tissue hormones, such as leptin or adiponectin. Lipogenesis is a term determining the synthesis of TAG from serum FFA or FFA derived from the hydrolysis of lipoproteins catalyzed by lipoprotein lipase (LPL). Insulin plays an important part in the process by stimulating the uptake of FFA by cells, activating lipogenic enzymes (such as acyl-coenzyme A (acyl-CoA) carboxylase, or fatty acid synthase), and inhibiting lipolytic enzymes, such as hormone-sensitive lipase (HSL). Activation of transcription factors, such as sterol regulatory element-binding proteins (SREBP) or peroxisome proliferator-activated receptors (PPAR), also stimulates this process. Lipogenesis refers also to de novo fatty acid synthesis in the cell cytoplasm. The opposite process—lipolysis—results in hydrolysis of TAG and the release of FFA from lipid droplets to the cytoplasm. The factors enhancing lipolysis are aforementioned HSL and also leptin, glucocorticosteroids, tumor necrosis factor α (TNF-α), and interleukin-6 (IL-6). Released FFAs are activated through binding to coenzyme A (CoA) and then transported into mitochondria to undergo β-oxidation. The summary of the homeostasis of FFA has been shown in Figure 1. FFA metabolism and the function of adipose tissue in FA homeostasis are further discussed in detail in numerous reviews [14,15,16,17].

The accumulation of FFA in non-fatty tissues may result in deleterious effects referred to as lipotoxicity. Dysfunction of adipose tissue fat storage ability, high-fat diet resulting in increased FFA plasma levels, and the shift towards lipolysis of TAG deliver an increased amount of FFA inside cells [21]. The other factor contributing to increased cytoplasmic FFA levels is the disturbance in perilipin expression and activity leading to either tissue steatosis or enhanced lipolysis delivering another load of FFA [14]. Cell metabolism requires efficient disposal mechanisms, such as esterification to TAG and intensified β-oxidation, to handle excess FFA [22,23,24]. In the case of non-adipose tissue, cell capabilities of FFA storage are limited and β-oxidation-required enzymes may become depleted. This leads to the accumulation of lipid metabolism intermediates, such as diacylglycerol (DAG) or ceramides, which can further disturb other metabolic pathways contributing to widely understood lipotoxic cell stress [25,26].

There is a positive correlation between TAG content and inflammatory responses [27] and insulin resistance [28], therefore suggesting that TAG depot is directly responsible for lipotoxic effects. However, the main cause of lipotoxic effects is the uncontrollable release of FFA from lipid droplets and adipose tissue and their interference with metabolic pathways, which also finds confirmation in studies [29,30,31].

## 3. Mechanisms of Lipotoxicity on the Cellular Level

### 3.1. Fatty Acids Metabolism

FFA are metabolized through a catabolic process referred to as β-oxidation which takes place in the mitochondrial matrix. The FFA intramitochondrial transport requires their activation to acyl-CoA esters in the outer mitochondrial membrane as well as carnitine transporter (carnitine palmitoyltransferase, CPT-1) which enables the transport of long and medium-chain fatty acids through the mitochondrial membrane in the form of acylcarnitines. Short-chain fatty acids do not require CPT-1 transportation. Β-oxidation consists of four biochemical reactions (oxidation, hydration, second oxidation, and thiolysis) repeated in cycles. During each cycle, the acyl-CoA ester is shortened by two carbon atoms, and acetyl-CoA, FADH2, and NADH are created. The cycle repeats, further shortening the acyl-CoA ester until the entire carbon chain is cleaved into acetyl-CoA. The produced acetyl-CoAs are fed into the citric acid cycle, whereas FADH2 and NADH are used in the electron transport chain. In brief, β-oxidation is a catabolic transformation of fatty acids into substrates for the creation of the electrochemical proton gradient and the production of ATP. The process of β-oxidation is further explained in other reviews [17,18].

Efficient β-oxidation is the key to maintaining proper FFA homeostasis in cells and preventing the accumulation of FFA in the cell, therefore preventing the lipotoxic effects. The importance of β-oxidation is supported by the observation that enhancing it protects against FFA-induced cytotoxicity. The enhancement of the β-oxidation was performed through the administration of CPT-1 stimulators (such as C75) into the cell culture and through in vivo electrotransfer of purified CPT-1 plasmid into rat tibialis anterior muscle to overexpress the muscle isoform of CPT-1. These actions resulted in the observation that the overexpression of CPT-1 and, therefore, an increased rate of β-oxidation improves insulin sensitivity in high-fat-fed rats and prevents FFA-induced oxidative stress, ER stress, apoptosis, and inflammation in neuronal and β-cell cultures [26]. The disturbance of FFA disposal mechanisms (inefficient or incomplete β-oxidation and incomplete esterification to TAG) results in abnormal accumulation of lipid metabolism intermediates, such as DAG and ceramides, as well as non-metabolized FFA, which can further interfere with cell signaling pathways, e.g., through the activation of protein kinases and, therefore, inhibiting insulin signal transduction [32]. Furthermore, palmitate (considered the most lipotoxic fatty acid) inhibits β-oxidation, creating a vicious circle [33,34]. There is a negative feedback mechanism, including malonyl-CoA, malonyl-CoA decarboxylase, and CPT-1. Malonyl-CoA is a product of carboxylation of acetyl-CoA and an allosteric inhibitor of CPT-1 resulting in the inhibition of FFA transport to mitochondria and subsequently β-oxidation. Inactivation of malonyl-CoA results in improved lipid oxidation and may prevent lipotoxicity [35], whereas the increase in malonyl-CoA level is inversely correlated to lipid oxidation.

The study was conducted by Asem et al. to determine whether changes in long-chain fatty acid (LCFA) oxidative metabolisms induced by high carbohydrate availability are a result of changes in LCFA mitochondrial transport capacity. The perfusion of 500 µM palmitate and [1-^14^C]palmitate or [1-^14^C]octanoate into rat hindquarters as well as with either low (LG) or high (HG) carbohydrate availability was performed. The postperfusion HG group malonyl-CoA levels were significantly higher than the LG group, and the relationships between percent and total palmitate oxidation and postperfusion muscle malonyl-CoA levels were hyperbolic and found to be significant [36]. Palmitate is observed to have an increasing effect on malonyl-CoA levels, which might explain its aforementioned β-oxidation inhibiting properties. Rat cardiomyocytes incubated in 500 µM palmitate showed an increase in malonyl-CoA level after chronic exposure (20 h) [37]. On the other hand, research shows that elevated plasma FFA in high-fat fed rats was associated with the increased expression of malonyl-CoA decarboxylase (MCD), which catalyzes the degradation of malonyl-CoA, creating feedback aimed at lowering FFA levels [38]. A similar situation to malonyl-CoA occurs with CPT-1, the expression and activity of which promotes β-oxidation; therefore, overexpression of CPT-1 enhances rat muscle insulin sensitivity and prevents FFA-induced apoptosis of murine podocyte cell [23,39], while its downregulation should exacerbate lipotoxicity. However, Haffar et al. showed that palmitate alone does not affect CPT-1 while impairing β-oxidation. The study was performed on rat neonatal cardiomyocytes, culture which was treated with 300 µM palmitate and 300 µM oleate for 8 h and the rates of β-oxidation, acetyl-CoA oxidation, citric acid cycle enzyme activities, CPT-1 activity, and DAG levels were measured. The results show that palmitate impairs β-oxidation and citric acid cycle flux, but no CPT-1 activity was observed. The impaired activity of the citric acid cycle enzymes may be due to DAG mediated protein kinase C (PKC) activation [33]. Another key regulator of β-oxidation is the 5′ adenosine monophosphate-activated protein kinase (AMPK), for which its activation triggers ATP production and inactivates an enzyme catalyzing malonyl-CoA formation—acetyl-CoA carboxylase (ACC). The activation of the AMPK pathway stimulates lipid oxidation and reduced murine podocytes’ susceptibility to toxic FFA effects by reducing palmitic acid-induced cell death [23]. Hickson-Bick et al. showed that acute 500 µM palmitate exposure causes a reduction in the activity of AMPK and an increase in the malonyl-CoA content; therefore, it decreases the cell’s capacity to oxidize fatty acids, resulting in the accumulation of FFA and lipotoxicity [37]. β-oxidation impairment has been observed in various diseases, such as obesity and insulin resistance in rat model, type 2 diabetes in humans, and renal injury in a mouse model, therefore suggesting that this branch of lipotoxic influences might be part of the etiology of those diseases [40,41,42].

Study shows that DAG, one of the intermediates of lipid metabolism, accumulates in the cell after acute exposure to increased FFA levels (300 µM of palmitate and oleate). According to Akoumi et al., DAG accumulates in the ER, and it is likely the key mechanism for FFA-mediated ER stress, which will be further discussed in Section 3.3 [26]. DAG is also recognized as a more lipotoxic agent than TAG. The research performed by Itani et al., Montell et al., and Chavez et al. on human muscle cultures and muscle cell cultures shows that the accumulation of DAG is mostly related to blocking insulin signaling and action, resulting in desensitization to the insulin stimulation of glucose uptake and further resulting in a condition referred to as insulin resistance [43,44,45,46]. It is hypothesized that the main mechanism of DAG’s influence on insulin signaling is due to its ability to activate PKC, a serine-threonine kinase, involved in various signal transduction pathways (such as MAP kinase signaling pathway or PI3K/Akt signaling pathway), for which its translocation to the cell membrane inhibits insulin signaling [47,48,49,50,51]. Due to the wide range of PKC activity, there was a strong premise that its activation is also responsible for FFA-mediated induction of the inflammatory pathways. However, research conducted by Macrae et al. excluded any direct involvement of DAG-sensitive PKC in palmitate-induced proinflammatory signaling [46].

Other intermediates generated by incomplete β-oxidation are acylcarnitines, which is the form of transportation for fatty acids through the mitochondrion membrane. Treatment of murine muscle cell culture with 10 µM of acylcarnitines for 24 h resulted in a 20–30% decrease in insulin response at the level of proteins kinase B (PKB, also known as Akt or Akt/PKB) phosphorylation and a 2–3-fold increase in oxidative stress. Another study shows that endogenous palmitoylcarnitine accumulation might contribute to palmitic-acid-induced insulin resistance in murine muscle cells [47,48].

Ceramides (sphingomyelin derivatives) are considered as potential lipotoxic molecules with the ability to modulate cellular metabolism. Palmitate, the most lipotoxic fatty acid, is activated to palmitoyl-CoA through esterification with CoA and can be transported by CPT-1 to the mitochondrion to undergo β-oxidation. However, palmitoyl-CoA can also serve as a substrate for ceramide synthesis via condensation with L-serine by the reaction catalyzed by an enzyme—serine palmitoyltransferase (SPT). Ceramides are signaling molecules that play a significant role in membrane biology (component of sphingomyelin, a major component of the lipid bilayer) as well as in several physiological events (such as regulating differentiation, proliferation, and programmed cell death). However, abnormal increases in ceramide content can contribute to toxic effects, such as suppressing muscle insulin action or apoptosis [49,50,51]. Since palmitate is an indirect substrate for ceramide synthesis, its increased levels in the cell have been shown to result in ceramide accumulation. Ceramide accumulation was observed in murine muscle cells after chronic incubation with 750 µM palmitate and in porcine oocytes after chronic incubation with 500 µM palmitate [45,52]. Moreover, the formation of ceramides via sphingosine acylation is observed to be a mechanism of lipotoxic action of palmitate on the cell, such as the induction of IR or apoptosis. Results from the study conducted by Manukyan et al. show the equal contribution of de novo and sphingosine pathways to ceramide generation in murine β-cells after chronic exposure to 250 µM palmitate [25]. Wehinger et al. proposed that the increase in ceramide levels caused by saturated FFA (16 h exposure to 2mM palmitate) induces mitochondrial damage and increases ROS generation and OS in cells, which in turn activates pro-apoptotic pathways in murine β-cells [53]. A study conducted by Chavez et al. presented that endogenously produced ceramides are capable of inhibiting Akt/PKB, a key element of the insulin signaling pathway, and are necessary for the inhibitory effect of FFA on insulin signaling. During the study, murine myotubes were incubated with 750 µM palmitate for 16 h [54]. These results are in accordance with Schmitz-Peiffer et al.’s observations that de novo ceramide synthesis mediates FFA inhibition of the PKB pathway in murine muscle cells [55]. Ceramides are also able to promote atypical isoenzyme PKCζ, which negatively regulates PKB by suppressing its cell–surface recruitment and phosphorylation as was measured after incubating rat myotubes with 750 µM palmitate for 16 h [56]. A novel study conducted by Sergi et al. on murine hypothalamic neuronal cells exposed acutely to 200 µM of palmitate explored the potential role of ceramides in FFA-induced hypothalamic neuronal inflammation. The results demonstrate that palmitic acid increases intracellular accumulation of ceramides together with the upregulation of IL-6 and TNF-α during treatment with the inhibitor of the key enzyme in the ceramide biosynthesis pathway, proving, at least, the partial involvement of ceramide in the mechanism of FFA-induced inflammation [57].

Palmitoyl-CoA, in addition to serving as a substrate for ceramide synthesis, also plays an important role in the process of protein palmitoylation. Protein palmitoylation is a process of post-translational protein modification in which palmitic acid is attached to the thiol group of specific cysteines. Numerous proteins serve as a target of palmitoylation, such as the following: CD36, G-protein-coupled receptors, ion channels and transporters, and others. Palmitoylation regulates protein localization, stability, and activity and increases protein affinity to the lipid rafts or cholesterol- and sphingolipid-rich membrane microdomains. Palmitic acid is present in the cell in the form of palmitoyl-CoA [58]. There is very little research concerning the process of palmitoylation in molecular mechanisms of lipotoxicity. However, dysregulation of the protein palmitoylation was proposed as one of the mechanisms by which palmitate may induce ER stress. Baldwin et al. showed that a nonmetabolizable palmitate analog that inhibits palmitoylation was able to attenuate palmitate-induced ER stress and cell death in rat β-cells. There was also a suggestion that uncontrolled protein palmitoylation is a foundation of palmitate toxicity [59]. Similar results were obtained in the study conducted on human neuroblastoma cells, where palmitate-induced ER stress, cell apoptosis, and also cell cycle G2/M arrest through protein palmitoylation were observed [60]. These results indicate that protein palmitoylation is a relevant aspect of lipotoxicity, which requires further research to elucidate new underlying mechanisms.

### 3.2. Oxidative Stress

Oxidative stress is a term referring to an imbalance between oxidizing and reducing agents with a shift towards oxidizing environment resulting from an increase in ROS content. The inability to detoxify ROS results in a free radical attack on proteins, lipids, carbohydrates, and nucleic acids, irreversibly altering them functionally or even destroying them completely. All organelles and compartments of the cells can produce ROS; however, the mitochondrial generation of hydrogen peroxide (H_2_O_2_) is generally considered to be a major source of oxidants, and additionally, H_2_O_2_ is highly reactive and acts on the biomolecules in its immediate environment [61].

Any disturbances in mitochondrion function are implications for a potential increase in ROS generation and lipotoxicity. An excess of FFA in the human hepatocytes (chronic exposure to 300 µM palmitate) and rat cardiomyocytes (chronic exposure to 250 µM palmitate up to 1 mM) can impair mitochondrial energetics, resulting in reduced ATP production followed by accelerated mitochondrial and cytosolic reactive oxygen species (ROS) production [62,63]. A few mechanisms of these deleterious FFA actions have been proposed. Szeto et al. proposed morphological changes in mitochondria structures leading to disturbances in electron transport chain (ETC) function and, subsequently, a decrease in ATP production in murine renal tissues [64]. Another mechanism could be a decrease in OXPHOS (oxidative phosphorylation system) enzyme activity, which is responsible for producing the majority of ATP, as was shown on human hepatocytes exposed to 200 µ palmitate for 24 h [65]. Decreased OXPHOS enzyme activity and reduced ATP production are consequences of lipotoxicity due to the uncoupling of these processes. FFA alone have uncoupling properties [66], and their interaction with mitochondrial carriers can also lead to membrane depolarization and the conversion of the carrier into a pore. Uncoupling proteins (UCPs) are members of the mitochondrial carrier family and their function is to regulate the proton’s gradient. There are several types of UCPs: UCP1 expressed in brown adipose tissue; ubiquitous UCP2; UCP3 expressed in skeletal muscle, heart, and brown adipose tissue; and others [67]. Research conducted on human hepatocytes shows that the inhibition of UCP2 enhances FFA-induced oxidative stress [68]. However, another research concerning isolated rat islets shows that, after chronic palmitate treatment, UCP2 expression significantly increased [69]. There is also an observed relation between UCP2 upregulation and autophagy in rat hepatocytes after palmitate treatment [70]. Those results suggest that interrelation between FFA and UCP is complex and reaches beyond mitochondrion function and can impair other branches of cell metabolism.

The progressive increases in mitochondrial DNA (mtDNA) damage after exposure to increasing concentrations of palmitic acid (most likely through the generation of mitochondrial ROS, mtROS) may also contribute to the dysfunctional mitochondria and further compromise the electron chain system, initiating the apoptotic machinery. This effect was observed in rat skeletal muscles after acute exposure to 500–2000 µM palmitate in bovine endothelial cells after chronic exposure to 50 µM palmitate in the presence of glucose and in rat skeletal muscles after chronic exposure to 1 mM palmitate [71,72,73]. FFA-induced ceramide accumulation (chronic incubation with 500 µM palmitate) in porcine oocytes was also observed to downregulate the AMPK/SIRT3 pathway and cause mitochondrial protein hyperacetylation, resulting in mitochondrial dysfunction [52]. Mitochondrial swelling was observed after chronic incubation of neonatal rat cardiac myocytes with 500 µM palmitate as a result of the opening of the mitochondrial permeability transition pore (MPTP), as manifested by cytochrome c (a component of ETC and apoptosis regulating factor) release to the cytosol [74].

Inhibition or deletion of cathepsin B, a lysosomal cysteine protease, protected against FFA-induced mitochondrial dysfunction in human hepatocytes, suggesting its role in the mechanism of FFA action on mitochondria as well as the role of lysosomes [75]. The role of cathepsin B finds confirmation in a study conducted on human hepatocyte cells, primary hepatocytes isolated from rats, and high-fat-fed rats by Wu et al. where the inhibition of cathepsin B expression and stabilization of lysosomal membrane was presented as major cellular mechanisms that accounted for 18β-glycyrrhetinic acid’s protective effects on FFA-induced lipotoxicity (chronic exposure 1 mM palmitate) [76]. However, in our view, the most abundant detrimental action FFA can execute on mitochondria is increased ROS production. This results in oxidative stress and also contributes to apoptosis, further damaging mtDNA and ETC [69,77].

FFAs are known to be one of the factors inducing the production of ROS. The family of NADPH oxidases (NOX) responsible for the transfer of electrons through biological membranes is a crucial factor in the production of ROS. There is evidence that FFAs (acute palmitate exposure) increase superoxide production through the activation of the NADPH oxidase system as well as by mitochondrial ETC in rat myocytes [77]. Another study conducted on human umbilical vein endothelial cells (HUVEC) exposed acutely to 1 mM palmitate suggests the possible mechanisms of FFA-induced NOX subunit gene expression [78] or the activation of NOX through PKC-dependent pathway in bovine aortic smooth muscle cells and endothelial cells treated with 200 µM palmitate for 72 h [79], thus contributing to increased ROS production. The NOX enzymes proven to be involved in lipotoxic ROS production are three isotypes members of the NOX family: NOX2, NOX3, and NOX4. NOX2 induction was observed in murine cardiomyocytes (acute 200 µM palmitate exposure), NOX3 was observed induction in human hepatocytes (chronic 250 µM palmitate exposure), and NOX4 induction was observed in human chondrocytes (chronic palmitate exposure, up to the concentration of 500 µM), suggesting tissue specificities of different isoforms [80,81,82].

Unresolved OS further damages the cell and can lead to apoptosis, as was shown on human hepatocytes treated with 100–800 µM palmitate for 24 h [83]. Another study conducted on third-order mesenteric arteries isolated from rats and HUVEC and treated with 100 µM palmitate for 24 h shows that lipotoxic OS can mediate vasodilatation via downregulation of potassium channels [84]. Researchers have distinguished various factors contributing to inducing lipotoxic oxidative stress. There is a strong link between oxidative stress and mitochondrial dysfunction since FFAs are involved in mtROS production and also decrease the expression of two major mitochondrial transcription factors, PGC-1α and mitochondrial transcription factor A (TFAM), as shown in a study conducted on rat skeletal myocytes treated with 100–1000 µM palmitate for 24 h [85]. The impairment of mitochondrial energetics was observed to be followed by accelerated mitochondrial and cytosolic ROS production in embryonic ventricular rat heart-derived cardiomyoblasts exposed to 250–1000 µM palmitate [63]. In high-fat-fed rats, palmitate stimulated oxidative stress in cardiac myoblasts and also altered protein levels involved in mitochondria functions: It increased the protein levels of CPT-1, cytochrome c, and cyclophilin F and induced the appearance of MPTP [86]. Mitochondrial oxygen consumption and ATP production are significantly decreased by FFA, resulting in increased mitochondrial proton leaks [87].

Yao et al. proposed another explanatory mechanism for FFA-induced mitochondrial dysfunction concerning the role of iron, which alone has little effect; however, when in combination with palmitic acid, it forms a complex and is shuttled across the lipid bilayer. Therefore, as observed in HUVEC treated with 300 µM palmitate in combination with 150 µM iron salt for 12 h, fatty acids mediate intracellular iron translocation, leading to iron overload in cytosolic and mitochondrial compartments that contribute to mitochondrial dysfunction, DNA mutation, and oxidative stress [88]. Fatty acids not only directly stimulate the synthesis of ROS but also disturb mitochondrial function, resulting in decreased antioxidant capabilities of the cell or indirect stimulation of ROS production. The aforementioned studies show that fatty acid-induced oxidative stress may be caused by suppressing β-oxidation and disrupting the mitochondrial respiratory chain, resulting in the accumulation of FFA in the cell. Accumulating oxidative damage may affect the efficiency of mitochondria and further promote ROS generation, therefore creating a vicious cycle [80,81,82].

One of the factors contributing to lipotoxic oxidative stress is the expression of fatty acid transporter CD36. In human kidney tissue, rat kidney tissue, podocyte, and human mesangial cells with increased expression of CD36 as well as increased lipid uptake, ROS production, and apoptosis were observed [89,90]. The novel study shows that perilipin 5 also has a crucial role in the alleviation of oxidative damage through the induction of antioxidant defense, which may suggest that stability of lipid droplets is an important aspect of maintaining redox balance [91]. Therefore, decreases in the activity of perilipin 5 or destabilization of lipid droplets are potential factors contributing to lipotoxicity. There is also a proposed model of FFA-induced cytotoxicity conducted on mouse podocytes, which provides evidence for close interaction between oxidative stress and Ca^2+^ homeostasis in mitochondria and the ER. The model states that FFAs are transported into the cell via CD36/FAT, resulting in an increase in mitochondrial and cytosolic ROS production. After the acute exposure of mouse podocytes to 300 µM palmitate, FFA-induced ROS production results in Ca^2+^ release from ER followed by ER Ca^2+^ depletion, ER stress resulting in podocyte apoptosis, and cytoskeletal derangement. The consequence of both podocyte apoptosis and cytoskeletal derangement is proteinuria. Another branch of the model links FFA-induced ROS production with mitochondrial dysfunction, which also contributes to podocyte apoptosis and proteinuria [92].

Aung et al. show that, in humans, brain microvascular endothelial cells (HBMECs) treated with isolated human triglyceride-rich lipoproteins (TGRL) oxidative stress associated with lipotoxicity are also connected to the induction of inflammatory pathways since the superoxide radical can activate activating transcription factor 3 (ATF3)-mediated inflammatory and apoptotic responses [93]. In a study conducted on cow endometrium epithelial cells treated chronically with 600 µM palmitate, Li et al. revealed that FFAs induce OS-mediated activation of the NK-κB signaling pathway, thereby leading to the release of inflammatory factors, such as IL-8, IL-6, and TNF-α [94]. Another research study conducted on rats infused with lipid emulsions by Barazzoni et al. pointed out FFA-induced ROS production, and the activation of the NF-κB pathway is also responsible for inducing IR, suggesting an interplay of various lipotoxic effects [95]. However, another research study conducted on isolated rat pancreatic islets treated chronically with 500 µM palmitate and other FFAs shows no immediate connection between OS and insulin resistance during the expression of the culture of β-cell islets in the presence of fatty acids [96]; therefore, this aspect requires additional research to determine and explain the relationship. A potential connection between oxidative stress and insulin resistance, two processes that can be induced by FFA, would suggest another interrelation of FFA-triggered lipotoxic molecular mechanisms.

Table 1 presents the sites of occurrence of lipotoxic oxidative stress molecular mechanisms. The table suggests that cardiomyocytes and myocytes are the most recurring sites of reviewed FFA-induced oxidative stress caused. There is further research and review required to determine whether those kinds of cells are the most prone to oxidative stress and whether this effect applies to more cell types. It also seems that lipotoxic OS is not tied to a specific cell type; however, confirmed lipotoxic OS occurs more often in cell types requiring increased metabolic activity for contraction, insulin secretion, and others. Another question that requires a confirmed answer is whether there are different susceptibilities to OS depending on cell type.

### 3.3. Endoplasmic Reticulum Stress

The phenomenon of ER stress concerns a large part of biochemical processes taking place in the cell. Therefore, the situation of unresolved ER stress triggers different signaling pathways leading to other disturbances. In the ER, proteins fold into their native conformation and undergo a multitude of post-translational modifications. Any disruption of these processes causes the accumulation of unfolded, aggregated proteins, which activate the unfolded protein response (UPR), an adaptive and protective mechanism aimed at restoring normal ER function. Three ER transmembrane protein sensors mediate UPR signals: IRE1 (inositol requiring enzyme 1; ERN1—ER to nucleus signaling 1), PERK (protein kinase RNA-like endoplasmic reticulum kinase), and ATF6 (transcription factor activating transcription factor 6) detect unfolded protein loads in the ER lumen and transmit signals to downstream effectors [97,98]. FFAs are one of the agents able to activate UPR. A study conducted by Nivala et al. on rats with administered lipid intravenous infusions shows that an increase in the amount of FFA may provoke ER stress and inflammation in liver and adipose tissues [99]. Another research study concerning mouse hepatocyte and murine podocyte cell cultures revealed that incubation with palmitic acid (chronic 600 µM palmitate treatment in human and rat hepatocytes and chronic 500 µM palmitate treatment in murine podocytes) leads to PERK and eukaryotic initiation factor 2 α (eIF2α) phosphorylation and increased CHOP (CCAAT/-enhancer-binding protein homologous protein, also known as C/EBP) expression that point out that ER stress was induced. It also elucidated the molecular interplay between signaling pathways involved in ER stress, insulin resistance, and apoptosis [100,101]. Unresolved FFA-mediated ER stress is responsible for triggering cell death via the activation of CHOP or caspase-12, or when ER stress exceeds the capacity of UPR in human hepatocytes (acute 1 mM palmitate treatment), in rat mesangial cells (chronic palmitate treatment) and rat myocytes (100 µM, 200 µM and 400 µM palmitate treatment for 12 h) [102,103,104]. There is also evidence that nuclear protein 1 (Nurp1), a transcriptional regulator downstream of CHOP, plays a critical role in apoptosis, as observed in human chondrocytes after chronic 500 µM palmitate treatment [105]. There is also the assumption that other factors, such as FoxO1, cannabinoid receptor 1 (CB1), or SREBP-1, can have a part in triggering apoptosis related to ER stress. Although the exact mechanisms have yet to be elucidated, there are observations that attenuating FoxO1 (observed in murine β-cells), CB1 (observed in human renal proximal tubular cells), and SREBP-1 (observed in a mouse model) activity prevents FFA-induced cell death [106,107,108].

The molecular chaperone BiP/GRP78 (binding immunoglobin protein/immunoglobulin heavy-chain binding protein) serves as a UPR regulator and is capable of activating all three transducers, IRE1, PERK, and ATF6, in response to ER stress. Under non-stressed conditions, BiP binds to IRE1, PERK, and ATF6, preventing their activation. When ER is overloaded with newly synthesized unfolded proteins, the pool of free BiP becomes depleted and then IRE1, PERK, and ATF6 are released from BiP, which enables signal transduction for UPR activation [98].

The IRE1 pathway regulates chaperone induction, clearance of unfolded proteins from the ER through upregulation of ER-associated degradation, and the expansion of the ER in response to ER stress to cope with the accumulation of unfolded or misfolded protein in the ER lumen. The activation of IRE1 results in alternative splicing and nuclear translocation of the transcription factor x-box binding protein 1 (XBP-1). Spliced XBP-1 mRNA encodes a potent transcriptional activator (an XBP-1 isoform) for many UPR target genes [97,98]. There is numerous evidence showing that FFAs induce ER stress through the activation of the IRE1 pathway. Although this pathway is meant to resolve ER stress, there is evidence that the activation of the IRE1 signaling pathway is involved in FFA-induced cell death of rat pancreatic β-cells (after chronic 1 mM palmitate treatment), porcine meniscus cells (after chronic 500 µM palmitate treatment), and neonatal rat cardiomyocytes (after acute 200 µM palmitate treatment) [109,110,111]. This effect seems to be tightly linked to inflammatory responses since the inhibition of FFA-induced activation of IRE1 abolishes IL-1β secretion stimulated by FFA, as observed in isolated primary mouse bone marrow-derived dendritic cells (BMDCs) [112].

Protein kinase mTOR (the mammalian target of rapamycin) complex 1 (mTORC1) (a master regulator in promoting growth and cellular anabolic processes in response to growth factors and nutrients excess) also plays an important role in the IRE1 pathway. Chronic 200 µM and 400 µM palmitate treatment is able to activate mTORC1 in murine hepatocytes; however, mTORC1′s inhibition attenuates palmitate-induced activations of IRE1, protecting cells against apoptosis; therefore, mTORC1 may be crucial for palmitate-induced ER stress-related cell death [113,114]. However, another study conducted on murine muscle cells provides data indicating that the activation of UPR represses the phosphorylation state of mTORC1, eventually resulting in a decrease in protein synthesis in the cell after chronic incubation with palmitate [115]. FFA can also partially induce XBP-1 mRNA splicing in murine preadipocytes (chronic palmitate treatment) and murine macrophages (chronic 400 µM palmitate treatment), causing less robust UPR than thapsigargin (a chemical ER stress inducer, which causes complete XBP-1 mRNA splicing) [116,117]. These observations show that FFAs interfere with the IRE1 pathway in multiple ways; however, further studies are necessary for determining whether the FFA-induced IRE1 pathway activation is dependent on cell type.

After activation by the release of BiP, PERK oligomerizes and phosphorylates substrate proteins, eIF2α, resulting in a reduced frequency of mRNA translation initiation in general. This results in the inhibition of the general protein biosynthesis, which serves as an adaptive mechanism in the accumulation of unfolded proteins [97,98]. Numerous studies conducted on two rat pancreatic β-cell cultures (after chronic 1 mM palmitate treatment in the case of the first cell culture and chronic 250 µM palmitate treatment in the case of the second cell culture) and human liver tissue (after chronic 600 µM palmitate treatment) show that the PERK signaling pathway is activated during FFA-induced ER stress, suggesting that FFAs are able to activate PERK [109,118,119]. Data also indicate that PERK and NF-κB signaling pathways, as well as JNK and PERK-dependent ATF3 activated by FFA, are involved in apoptosis through the regulation of the expression of Bcl-2 family members, as observed in human hepatocytes and human and rat β-cells after chronic treatment with FFA [120,121]. Studies conducted on murine preadipocytes and human islets (after chronic palmitate treatment in both cases) show that palmitate-induced ER stress-related apoptosis is associated with the increase in transcription factor phospho-eiF2α [116,122]. The direct effect of these actions is a decrease in protein synthesis, which also has been observed after chronic FFA exposure in murine muscle cells [115]. Even though the decrease in protein synthesis is meant to rescue ER protein-folding abilities, this mechanism is not sufficient for resolving ER stress induced by FFA, and the final result is often apoptosis as observed in human hepatocytes, rat mesangial cells, and rat myocytes [102,103,104]. Upon ER stress, phosphorylated eiF2α selectively promotes the translation of activating transcription factor 4 (ATF4) mRNA. ATF4 is required for the expression of genes involved in amino acid import, glutathione biosynthesis, and resistance to oxidative stress. It binds to CHOP, GADD34 (DNA damage-inducible protein), and ATF3. CHOP forms heterodimers with C/EBP family members and controls the expression of genes involved in apoptosis [97,98]. Anusornvongchai et al. and Cao et al. showed that both ATF4 mRNA expression and ATF4 protein expression were increased in human hepatocytes chronically exposed to FFA, therefore suggesting that FFAs are capable of inducing the expression of ATF4 [123,124]; however, it is not certain whether this phenomenon is an independent action or an indirect result of PERK activation. The ability of FFA to influence regulatory elements of UPR refers also to observed increased expressions of CHOP, CHOP gene, and CHOP mRNA under the influence of FFA in rat β-cells, human and mouse neuroblastoma cells, and bovine mammary epithelial cells after chronic exposure to palmitate treatment [125,126,127].

Different effects concerning FFA-induced ER stress have been observed in studies conducted on murine β-cells after chronic palmitate treatment, such as disrupted ER-to-Golgi protein trafficking resulting in protein overload in contrast to protein-synthesis inhibiting properties [128,129]. These observations are a foundation of the concept that protein overload, rather than disruption of the protein-folding capacity of the ER, is an underlying mechanism of FFA-induced ER stress in β-cells. Another mechanism concerning the depletion of the ER Ca^2+^ reservoir has been proposed as a mechanism of FFA-induced ER stress. According to Cunha et al., chronic saturated FFA treatment decreases ER Ca^2+^ stores in rat insulin-producing β-cell culture and ER Ca^2+^ depletion activates ER stress [130]. The rapid depletion of ER Ca^2+^ in response to FFA was also observed in mouse insulinoma-derived cells [131]. Zhang et al. suggested that the flux of calcium is required to mediate FFA-induced ER stress, as observed in rat hepatocytes after chronic palmitate exposure [132]. Research data suggest that ER calcium depletion is connected with dysregulated mitochondrial metabolism and that accelerating mitochondrial Ca^2+^ clearance might relieve cytosolic Ca^2+^ overload in rat hepatocytes after chronic treatment with 400 µM palmitate and mouse β-cells after chronic treatment with 500 µM palmitate [133,134]. Ca^2+^ flux is also associated with cell death. In mouse β-cells, Ca^2+^ activated phosphatase calcineurin was suggested to be involved in apoptotic processes via the dephosphorylation of apoptosis-related molecules such as Akt and b-cell lymphoma-2(Bcl-2)-associated death promoter (Bad) [135]. Another molecule with a possible contribution to Ca^2+^-related apoptosis is calpain-2, a calcium-dependent proapoptotic protease. Calpain-2 was activated in rat insulinoma cells, mouse islets, and human islets treated for 24 h with 500 µM palmitate, suggesting that ER Ca^2+^ depletion can result in the activation of calcium-dependent cell death pathways in the cytoplasm [136].

The ER stress promotes IL-1β synthesis through the stimulation of NF-κB pathways and stimulates inflammasome-dependent responses, therefore contributing to activating inflammatory responses in the cell, as observed in human leukemic monocytes, murine macrophages, and murine adipocytes [137]. Kim et al. demonstrated that FFA-induced ER stress in human vascular endotheliums is mediated through toll-like receptor 4 (TLR4) and that the reduction in ER stress restores vasodilator actions of insulin [138]. There is another visible connection between ER stress and OS. Both these processes contribute to FFA-induced cell death and their signaling pathways might be related, as observed in mouse podocytes, human osteoblast-like cells, and rat renal proximal tubular cells [92,139,140]. There is a probability that both ER stress and OS trigger each other; however, it is also possible that these processes occur simultaneously because both are induced by FFA. More study is needed to elucidate this connection.

Lipotoxicity and subsequent FFA-induced ER stress is also proposed as a mechanism underlying several diseases. Ozcan et al. conducted research on rat hepatocytes and a mouse model and identified lipotoxic ER stress as a molecular link between obesity, the deterioration of insulin action, and the development of type 2 diabetes [141]. The same applies to the development of non-alcoholic fatty liver disease (NAFLD), as observed in rat and mouse in vivo models [142,143]. Another study demonstrated that triggering ER stress might be responsible for lipotoxic rat myocardial injuries [104] and renal proximal tubule injuries in mice [108]. However, due to the differences between experimental FFA administration and pathological disturbances in FFA homeostasis resulting from disease, further studies are needed to confirm lipotoxic mechanisms as mechanisms underlying the mentioned diseases. There also are findings suggesting a connection between FFA-induced ER stress, insulin resistance, and inflammation. Ebersbach-Silva et al. pointed out that FFA-induced reductions in glucose transporter type 4 (GLUT4) expression in rat muscle cells after acute FFA exposure is related to the formation of an IRE1α-TNF-associated factor 2-IκB kinase (IRE1α-TRAF2-IKK) complex and the activation of NF-κB, therefore linking the mechanisms of FFA-induced IR, inflammation, and ER stress [144]. Another study conducted on isolated mouse pancreatic acinar cells indicates that, in FFA-stimulated pancreatic acinar cells, ER stress induces inflammatory responses through the induction of the C/EBP family [145]. Those observations might confirm the vicious cycle of lipotoxicity and interplay between signaling pathways involved in processes induced by FFA, such as ER stress, IR, or inflammation.

Table 2 demonstrates lipotoxic molecular mechanisms of ER stress and their sites of occurrence. The cell types that stand out as the most frequent site of lipotoxic ER stress are hepatocytes and pancreatic β-cells; however, other cell types are also included. It seems that most mechanisms are present subsequently in the mentioned cell types with the exception of mTORC1 activation, which seems to be tied to the liver based on the current review. Moreover, perturbations of protein trafficking and ER Ca^2+^ depletion are mainly observed in pancreatic β-cells, suggesting the cell-type specificity of those mechanisms, probably due to specific functions of β-cells, such as insulin secretion.

### 3.4. Inflammatory State

Chronic FFA treatments have been proven to increase cell-autonomous inflammatory responses in various ways, e.g., stimulation of pro-inflammatory gene expression or triggering inflammatory signaling pathways, as observed in human coronary artery smooth muscle cells (HCASMC) and human coronary artery endothelial cells (hCAECs) [147,148]. Another method of activating inflammatory responses is triggering other dysfunctions that further promote inflammation, such as mitochondrial dysfunction, overproduction of ROS, induction of C-reactive protein gene expression, and the following suppression of NO production, as shown in HBMEC treated with isolated human TGRL, rat and human myotubes, and human aortic endothelial cells (HAECs) [93,149,150].

FFA can also induce systemic inflammatory responses tied to adipose tissue through the activation of macrophages or chemotaxis. Macrophages, the primary mediator of the immune response, are present in adipose tissues in lean subjects and their amount is directly proportional to adiposity [151]. We can distinguish two major subpopulations of macrophages: inflammatory “classically activated” M1, which produces proinflammatory cytokines, and anti-inflammatory “alternatively activated” M2, which produces anti-inflammatory cytokines. In lean, healthy subjects, resident macrophages express markers of a subpopulation of M2. Obesity, i.e., excess of white adipose tissue (WAT), imposes a new population of M1-polarized macrophages on resident ones. The balance of M1/M2 macrophages is then shifted towards a more pro-inflammatory environment in adipose tissue, including a high expression of TNF-α, IL-6, and inducible nitric oxide synthase (iNOS) [152,153]. The chemokine system is understood as a chemokine receptor and its ligand (e.g., C-C chemokine receptor type 2 (CCR2) and monocyte chemoattractant protein MCP-1) plays a crucial role in obesity-induced adipose tissue inflammation via macrophage recruitment, i.e., increased infiltration of the adipose tissue with macrophages [153]. There is evidence that FFAs are able to induce local secretions of chemokines, therefore provoking macrophage infiltration. Induced expressions of MCP-1 was observed in mouse myotubes treated for 24 h with palmitate, and conditioned media from the palmitate treatment mouse myotubes displayed enhanced properties for recruiting mouse macrophages [154]. Similarly, chemotaxis of murine macrophages towards murine β-cells following treatment with conditioned media from murine β-cells treated with palmitate for 24 h was significantly increased [155]. Induced secretions of MCP-1 was also observed in murine adipocytes chronically treated with palmitate [156].

The goal of recruitment of M2 macrophages is to maintain the lipid homeostatic state within WAT by means of β-oxidation of excess FFA. The inability of these macrophages to help relieve excess FFA within WAT contributes to systemic lipid dysregulation and chronic metabolic-related systemic inflammation [157,158], and FFA themselves are proven to activate M2 macrophage polarization, as observed in chronically palmitate-treated murine macrophages [159]. Based on the studies on rat myoblasts and murine macrophages, it is suggested that FFAs are able to induce macrophages to secrete proinflammatory cytokines (TNF-α and IL-6), augmenting the inflammatory response, through c-Jun N-terminal kinase (JNK) and p38 Ras-mitogen-activated protein kinase (MAPK) phosphorylation [160]. Proinflammatory macrophages can worsen or impose the lipotoxic effects on other cells by infiltrating tissues and releasing proinflammatory cytokines in a mouse model [161,162]. However, macrophages can also suffer from an excess FFA in their cells and undergo lipotoxic ER stress or apoptosis, since these processes are not tissue-specific.

TLRs are innate immune cell receptors that include several subtypes (e.g., TLR1, TLR2, TLR3, TLR4, etc.) and it was believed that they can be activated by FFA. TLRs recognize pathogen-associated molecular patterns (PAMPs) and damage-associated molecular patterns (DAMPs), exogenous and endogenous molecules promoting cell-autonomous inflammatory responses, and subsequently activate intracellular signaling pathways via the adaptor molecule myeloid differentiation primary response 88 (MyD88). MyD88-dependent signaling triggers the classical inflammatory cascade, resulting in NF-κB and MAP kinase activation [163,164]. The interaction between TLR and MyD88 results in autophosphorylation of IL-1R-associated kinase (IRAK). Activated IRAK interacts with TNF-associated factor 6 (TRAF6), resulting in the stimulation of JNK and IKK (inhibitor of a κ-B kinase; prevents NF-κB inhibition) signaling pathways, which directly contribute to inducing inflammatory responses [165,166].

There is numerous evidence that FFAs participate in the activation of TLR1, TLR2, and TLR4. TLR1 and TLR2 are activated by receptor dimerization, resulting in the activation of downstream signaling pathways and target gene expression, as stated in the study conducted on murine macrophages and human monocytes. As a result, inflammasome-mediated IL-1β production is induced [167,168]. There are also findings in murine macrophages and rat β-cells that lipotoxic activations of TLR4 are involved in FFA-induced cell death through the activation of the JNK pathway or intersection between TIR-domain-containing adapter-inducing interferon-β (TRIF) signaling and impaired lysosome function [164,166]. In addition to the JNK pathway, the activation of Blc-2-associated X protein (Bax), a pro-apoptotic member of the Blc-2 family, and the engagement of the mitochondrial proapoptotic pathways are observed, as observed in human hepatocytes chronically treated with palmitate and high-fat fed rats [169,170,171]. The imbalance in proapoptotic (e.g., Bax and Bak) and antiapoptotic (e.g., Bcl-2) protein in the Bcl-2 family was also highlighted by research conducted on human hepatocytes treated for 8 h with 800 µM palmitate and mouse islet chronically treated with 500 µM palmitate by Akazawa et al. and Litwak et al. [172,173]. Proapoptotic proteins of the Bcl-2 family can induce the release of cytochrome c from mitochondria to the cytoplasm, which activates the caspase-dependent apoptotic signaling pathway. There are observations that FFA intervention results in increased expressions of caspases 3 and 9 (key enzymes of the mitochondrial apoptotic pathway) and decreased expressions of antiapoptotic factor Bcl-2, which carries the potential to alleviate FFA proapoptotic actions. FFA can also significantly induce cytochrome c release in mouse β-cells and human lymphocytes [174,175]. Studies in mouse macrophages and mouse myocytes also provided a link between the activation of TLR2 and TLR4 by FFA and FFA-induced IR by the activation of macrophages via TLR2-, TLR4-, and JNK-dependent inflammatory pathways and the activation of TLR2 in muscle cells [176,177]. As observed in rat β-cells and murine microglial cells, the potential phenomenon of activating TLR gives FFA the ability to stimulate signaling pathways of JNK and NF-κB [166,178], which results in an increased production of pro-inflammatory cytokines and sustained inflammation. Although there is a significant difference between experimental FFA treatment of the cell culture, the activation of TLR is a proposed mechanism of non-alcoholic steatohepatitis (NASH) development from NAFLD in a mouse model [179]. Therefore, this phenomenon might serve as a beneficial beacon for future studies.

The indirect cell-autonomous inflammatory effects of FFA are also related to other lipotoxic actions, such as the aforementioned increased production of ceramides, DAG, or ROS. DAG is an allosteric activator of PKC isoforms. An excess of FFA and following a rise in DAG amount causes the activation of PKC, which can induce OS and activate NF-κB. The activation of the PKC-NF-κB axis and subsequent increase in IL-6 expression has been proposed as an FFA-induced mechanism of insulin resistance based on human studies and studies on mouse myoblasts [43,180]. Increased levels of ceramides are not only an effect of lipotoxicity caused by FFA but are also induced by augmented signaling through TLR4. Ceramides inhibit glucose uptake by blocking the activation of Akt/PKB, which contributes strongly to insulin resistance, as observed in lipid infused mice [181]. ROSs are able to drive ATF3-mediated inflammatory responses in HBMEC treated with isolated human TGRL [93], and FFAs are known for stimulating the synthesis of ROS; thus, the conclusion is that inducing OS by FFA is associated with inducing inflammation. There was also an observation that, in murine macrophages, chronic FFA exposure can induce the expression of cyclooxygenase-2 (COX-2), an important enzyme in prostaglandin synthesis from arachidonic acid, which is expressed in the process of inflammation [182]. Based on studies conducted on rat muscle cells treated acutely with FFA, the formation of the IRE1-TRAF2-IKK complex (with another type of TRAF than mentioned before) and the activation of NF-κB has also been linked to the palmitate’s ability to reduce GLUT4 gene expression, therefore indicating the connection between lipotoxic ER stress, inflammation, and insulin resistance [144]. The subsequent presence of lipotoxic inflammation and insulin resistance has been observed in rat myotubes treated chronically with palmitate [149].

Findings confirm that IRE1, the ER stress sensor, can be activated via TRL, leading to the production of pro-inflammatory cytokines in a mouse model and mouse macrophages [183,184,185]. This phenomenon means that FFA activating IRE1 and TLR simultaneously also triggers TRL-mediated activations of IRE1, leading to the development of both ER stress and inflammation. Moreover, FFA-induced activation of IRE1 can mediate the activation of NOD-like receptor family pyrin domain-containing 3 (NLRP3) inflammasome, an intracellular protein complex that assembles in response to DAMP and PAMP and catalyzes the cleavage and maturation of cytokines IL-β1 and IL-18. Such observations were made in isolated primary mouse BMDC, a mouse model of long-term high-fat fed mice, mouse hepatocytes, and human studies [112,186,187]. FFAs act as a DAMP to activate the NLRP3 inflammasome in the NASH mouse model. NLRP3 inflammasome activation can also be mediated by sensing mitochondrial dysfunction (through the direct binding of NLRP3 with mtDNA) and the decrease in mitochondrial membrane potential, possibly an important prerequisite of mtDNA release from the mitochondria to the cytoplasm (a key factor in NLRP3 inflammasome activation), as observed in Kupffer cells isolated from mouse livers [188]. The study conducted by Weber et al. shows that lysosomes play an important role in the lipotoxic NLRP3 inflammasome in primary mouse macrophages. In the cell treated with both palmitate and lipopolysaccharide (LPS), lysosome destabilization contributed to NLRP3 and caspase-1 activation and subsequent IL-1β release [189]. FFA can also induce inflammasome-mediated IL-1β production in human monocytes by inducing the activation of TLR2 [168].

Due to the activation of various cell-autonomous inflammatory responses, FFA stimulates the secretion of pro-inflammatory cytokines such as TNF-α, which further induces inflammatory pathways such as IKKβ/NF-κB or PKC/NF-κB cascades. Such observations were made in lipid-infused and high-fat fed mice, in mouse myotubes treated chronically with 500 µM palmitate, and in human hepatocytes treated chronically with 500 µm and 1 mM FFA (2:1 oleate/palmitate) [190,191,192]. FFA also induce sensitization to TNF-related apoptosis-inducing ligand (TRAIL) mediated apoptosis, as observed in human hepatocytes treated chronically with a mixture of stearic acid and oleic acid with the addition of recombinant human TRAIL [193]. The aforementioned IL-1β subsequently activates IL-1R, which results in IL-1β auto stimulation and the further upregulation of IL-1-dependent chemokines and cytokines, such as IL-8 and IL-6. IL-1R mRNA expression was significantly increased in human islets after chronic exposure to 500 µM of palmitate, stearate, and oleate separately [194].

The means of FFA-associated activation of inflammatory responses are so numerous and intertwined that they resemble a network rather than a straight pathway. The complexity of this network means that FFAs have several means (both direct and indirect) of inducing the same signaling pathways and that FFA-induced inflammation cannot be stopped with the attenuation of one transmitter.

Table 3 presents summarized molecular mechanisms of inflammatory responses induced by FFA. Even though most of those mechanisms concern cells of the immune system, there are examples of inflammatory responses in different types of cells that are not tied to the immune system. Macrophages activated towards inflammation may contribute to systemic inflammation and infiltration of tissues invoking local inflammatory response and worsening further pathological states, such as IR, ER stress, and OS.

### 3.5. Induction of Insulin Resistance

Insulin resistance alone is considered a disease, but it can also lead to the development of T2DM or metabolic syndrome when untreated. FFAs have been proven to induce insulin resistance and several aspects and mechanisms of this action have been proposed. One of them is the inhibition of glucose transport and phosphorylation with a subsequent reduction in rates of glucose oxidation and glycogen synthesis. Due to defects in β-oxidation, fatty acid metabolites, such as acyl-CoA, are generated, with an ability to induce a defect in glucose transport. Acyl-CoA inhibits the activity of enzymes responsible for the metabolism of glucose-6-phosphate and accumulates glucose-6-phosphate, which inhibits glucose transport by negative feedback [196,197]. FFAs also might inhibit the insulin suppression of endogenous glucose production by interfering with the insulin suppression of glycogenolysis, as observed in human studies [198]. Impaired glucose metabolism is also linked to mitochondrial dysfunction, thus suggesting that FFA-induced mitochondrial dysfunction (e.g., disruption of the mitochondria-associated ER membrane) could be an early event in the development of FFA-induced IR. The reason for mitochondrial dysfunction is most likely an inability to match oxidative capacities to excess required ATP content. The link between impaired metabolism and mitochondrial dysfunction was demonstrated in studies on rat and mouse myotubes treated chronically with 100 µM palmitate and human hepatocytes (insulin resistance induced after chronic 300 µM palmitate treatment and mitochondrial dysfunction induced after acute 300 µM palmitate treatment) [149,199,200]. Therefore, the accelerated transport of FFA to the mitochondria and the improvement of mitochondrial fatty acid β-oxidation seem to protect the cell from FFA-induced IR, as observed in rat skeletal muscle cell treated chronically with 250 µM palmitate and mouse myocytes treated chronically with 500 µM palmitate [201,202,203].

Another mechanism suggested to be responsible for FFA-induced IR is the inhibition of insulin signaling. In the presence of insulin, the insulin receptor (INSR) phosphorylates the insulin receptor substrate proteins (IRS proteins). The activation of the two main signaling pathways occurs: the phosphatidylinositol 3-kinase (PI3K)-Akt/PKB pathway (responsible for most of the metabolic actions of insulin) and MAPK pathway (regulating the expression of some genes and cooperating with the PI3K pathway) [204]. FFAs can alter insulin signaling through the phosphorylation of IRS proteins, as demonstrated in rat hepatocytes after acute 200 µM palmitate treatment, human podocytes after chronic 750 µM palmitate treatment, and human skeletal muscle cells after chronic 400 µM palmitate treatment [205,206,207]. The activities of various elements of insulin signaling pathways, such as PKB or PI3K kinase, have been observed to be altered by FFAs in human studies and in a rat model subjected to lipid infusion [208,209]. The coexisting inhibition of insulin signaling-associated gene expression also has been related to the lipotoxic action of chronic FFA treatment of human podocytes and rat hepatocytes [206,210].

Further research on the mechanism of IR caused by FFA resulted in observations that lipotoxic metabolic fatty acid intermediates (DAG, ceramides, and acylcarnitines) disturb insulin action. IR is correlated with the accumulation of DAG, ceramides, and acylcarnitines, which is a result of the inhibition of mitochondrial FFA uptake, incomplete β-oxidation, and others (e.g., de novo ceramide synthesis) [45,47,49]. Some researchers conducted studies on human podocytes and high-fat fed rats and proposed that accumulated FFA intermediates not only correlate with IR but are responsible for its induction [206,211]. The inhibition of PKB signaling was proven to be derived from elevated levels of ceramide by promoting activations of an atypical isozyme of PKC, PKCζ, which negatively regulates PKB [56]. Furthermore, the ability of DAG to activate PKC plays a crucial role in FFA-induced IR. The development of IR in lipotoxic conditions (human studies and rat NAFLD model) is associated with an increase in PKC activity, suggesting that PKC-activating action of DAG is at least in part involved in the mechanism of FFA-mediated IR induction [43,212,213]. This feature resulted in assuming DAG to be a more important determinant of IR than ceramide, as observed in a study on rat skeletal muscle cells after chronic palmitate treatment [214]. In studies on lipid infused rats and rat skeletal muscle cells, the activation of various isoforms of PKC has been linked to disturbed insulin signaling as a result of, e.g., decreased activation of IRS-1-associated PI3 kinase [215,216,217]. However, there is research conducted on mice by Monetti et al., contradicting the role of DAG in IR, where accumulated lipotoxic intermediates did not cause IR, which resulted in an assumption that other mechanisms and metabolic pathways, such as inflammation, may be required for the development of IR [218].

There are data from studies on rat myoblasts and murine macrophages, as well as human hepatocytes and murine macrophages, indicating that IR is associated with inflammation and suggesting that FFA-induced local secretion of MCP-1 participates in the recruitment of macrophages. Moreover, FFAs mediate M1 macrophage polarization, therefore creating an inflammatory environment through the release of proinflammatory cytokines [160,219]. The linkage between FFA, inflammation, and IR is also demonstrated through the activation of a master regulatory signaling pathway of inflammation—NF-κB or decreased IκB, an inhibitor of NF-κB, as observed in human studies and murine myoblasts after chronic 1 mM palmitate treatment [43,220]. In a study performed on rats infused with lipid emulsions, Barazzoni et al. pointed out that the activation of NF-κB is mediated by the enhancement of ROS production by fatty acids, therefore indicating ROS participation in IR induction [95]. Another study on rat hepatocytes treated chronically with palmitate shows that ROS has a part in desensitizing insulin signaling pathways by activating JNK, impairing the phosphorylation of IRS [221]. There are also reports from studies on human hepatocytes (chronic 250 µM palmitate exposure) and on high-fat fed mice that NOX2 and NOX3 may contribute to IR by mediating ROS generation and that inhibiting the activity of NOX oxidases may restore insulin sensitivity [80,222]. This suggests the connection between two FFA-induced processes: ROS and IR. Other than that, ER stress may also have a role in the development of IR, as observed in murine mesentery arterioles after intraluminal palmitate infusion and in mice in vivo studies [138,223].

There is a study on murine and human myotubes treated chronically with various concentrations of palmitate (250 µM, 500 µM, and 750 µM) that showed that ER stress does not mediate IR, but a marked activation of UPR can induce IR at the level of IRS through the activation of the IRE-1/JNK pathway [224]. The matter of ER stress involvement is, therefore, debatable and needs further research to determine actual linkage; however, upon the aforementioned observations, there seems to be a connection of unknown nature between ER stress and IR, and we cannot state for sure that these processes are completely separate. There is evidence that the activation of TLR2 mediates proinflammatory responses that contribute to IR with the participation of MyD88, JNK, and NF-κB, as observed in mouse myocytes (acute and chronic 750 µM palmitate treatment), HAEC (acute 200 µM palmitate treatment), and murine preadipocytes (chronic 500 µM palmitate treatment) [177,225,226]. The fact worth mentioning is that the activation of TRL2 also mediates UPR, and that is another link potentially connecting ER stress with FA-induced IR.

FFAs are putative mediators of β-cell dysfunction, which may lead to the loss of functional β-cell mass and, therefore, contribute further to IR and the pathogenesis of T2DM. As demonstrated in human studies and in rat β-cells incubated chronically with 200 µM palmitate and glucose, FFAs inhibit insulin biosynthesis, leaving β-cells drained from their islet insulin content and with limited means to replenish it [227,228]. The disruption of insulin biosynthesis might occur due to the direct inhibitory effect of FFA on insulin gene expression. There are reports from studies on rat β-cells (chronic palmitate treatment), isolated rat islets (chronic palmitate treatment), and hamster β-cells (chronic palmitate treatment) that this effect takes place only in the presence of high glucose [229,230,231]. The mechanism behind this phenomenon is not yet elucidated, but there might be involvements of JNK activation (mouse hepatocytes after acute 500 µM palmitate treatment) [232], inhibition of insulin promoter activity, and de novo ceramide synthesis (rat islets after chronic 500 µM palmitate treatment) [233], or reduced binding of key regulatory factors (transcription factor MafA and pancreas-duodenum homeobox-1 (PDX-1)) to the insulin gene promoter (rat islets after chronic 500 µM palmitate treatment) [234]. As a consequence of decreased insulin gene expression, intensified lipogenesis is present, resulting in increased DAG and TAG content [231].

In addition to serving as an energy substrate, FFAs can also act as signaling molecules interacting with FFA receptors (FFARs), a family of G protein-coupled receptors (GPCRs). Currently, five types of GPCRs are known to be activated by FFA: FFAR1, FFAR2, FFAR3, FFAR4, and GPR84. These receptors are expressed in numerous sites, such as β-cells, immune cells, central nervous system, or adipose tissue, and they play a part in insulin secretion, inflammatory responses, and other significant signaling pathways [235]. Although the role of FFAR in cell metabolism is well described, their role in the molecular mechanisms of lipotoxicity requires further research. The majority of research concerning FFAR and lipotoxicity refers to FFAR1′s role in β-cells. A study conducted by Marafie et al. demonstrated a possible regulatory role for FFAR1 in insulin signaling in rat β-cells under lipotoxic conditions (chronic palmitate treatment) [236]. Another study confirms that FFAR1 mediates, in part, the palmitate-induced (chronic palmitate treatment) potentiation of GSIS in mouse β-cells. Furthermore, inhibition and activation of FFAR1 seem to impose a different effect on the cell depending on FFA levels [237]. Graciano et al. showed that FFA-stimulated (acute palmitate treatment) superoxide production and insulin secretion are FFAR1-dependent in rat β-cells [238]. The level of FFAR1 expression is also negatively correlated with β-cell apoptosis. FFAR1 also plays a role in pioglitazone-mediated attenuation of FFA-induced oxidative stress and apoptosis, as observed in mouse β-cells after chronic palmitate treatment [239]. Similarly, exendin-4 prevents the proapoptotic effect of palmitate by reducing FFAR1 expression in human and murine pancreatic islets and rat β-cells [240]. Therefore, FFAR1 might serve as a direction of research on pharmacological agents with the ability to attenuate lipotoxicity in β-cells.

There also is a concept of glucolipotoxicity concerning β-cells, which includes the combined deleterious effect of both high FFA and glucose levels. The term was proposed to describe FFA lipotoxicity occurring in β-cells, which in this case is often dependent on high glucose levels, resulting in OS, loss of insulin secretory function, β-cell death, and inflammation, as observed in rat pancreatic islets and mouse and rat β-cells [241,242].

### 3.6. Autophagy

Autophagy is a process of cell self-digestion that regulates cellular component degradation through lysosomes by degrading malfunctioning organelles and proteins and plays an important role in maintaining cell homeostasis by regulating the synthesis and degradation of cellular components. Various data indicate that autophagy serves as an adaptive process that could overcome the deleterious effect of FFA on the cell, as observed in rat β-cells after chronic 400 µM palmitate treatment and in murine adipocytes after chronic treatment with 500 µM and 1 mM palmitate [243,244]. In physiological conditions, this process is upregulated by the deprivation of nutrients and growth factors.

There are many observations that excess FFA is able to increase the rate of autophagy concerning high-fat fed mice, mouse hypothalamic neuronal cell line, rat β-cells (after chronic palmitate exposure), and isolated rat pancreatic islets (after chronic palmitate exposure) [245,246,247]. A study on rat β-cells after chronic 400 µM palmitate treatment suggests that increased autophagy could overcome FFA-induced cell death and delay cell loss [243]. Evidence shows that the activation of autophagy under lipotoxic conditions occurs in a JNK-dependent manner, as observed in murine adipocytes after chronic treatment with palmitate, rat β-cells after acute treatment with 500 µM palmitate, and murine β-cells after chronic treatment with 500 µM palmitate [244,248,249]. It has been shown earlier that FFAs as TLR agonists are able to activate the JNK signaling pathway; therefore, it is possible that the activation of autophagy is a direct effect of saturated fatty acids. Another study on high-fat fed mice suggests that ceramides have their part in lipid-induced autophagy, pointing out that FFA could have imposed an indirect autophagy-inducing effect [250]. Additionally, Tan et al. indicated that FFA promoted DAG accumulation, and subsequent PKC activation is an upstream signaling mechanism for autophagy in mouse embryonic fibroblasts treated acutely with 250 µM palmitate [251].

The aspect of dependency of FFA-induced autophagy on other FFA-induced processes, such as ER stress or OS, is debatable. Quan et al. pointed out that autophagy is necessary for proper UPR action, leading to an assumption that both autophagy and ER stress are related processes, as observed in murine β-cells after chronic palmitate treatment [252]. Dependency of autophagy on ER stress was observed in another study on human osteoblast-like cells [139]. Yin et al. suggested that autophagy is a result of cellular stress induced by FFA, after the observation made on murine adipocytes after chronic treatment with palmitate that autophagy is induced subsequently to the activation of the ER stress-JNK pathway and plays a protective role against FFA-induced cell death, confirming the relationship between the processes [244]. Furthermore, Chen et al. suggested that FFA-induced autophagy activates Ca^2+^/PKCα/NOX4 pathways to promoter ROS generation, as observed in HUVEC after chronic 300 µM palmitate treatment [253]. However, a different study conducted on rat β-cells after acute treatment with 500 µM palmitate demonstrated that FFA-stimulated autophagy occurs independently of oxidative or ER stress [248]. Further research is needed to determine the essence of relations between the processes.

There are many studies showing that FFAs are able to inhibit autophagy, worsening their detrimental impact. Nutrient abundance has been associated with the suppression of autophagic turnover, resulting in the accumulation of autophagosomes, autophagy substrates, ER stress, and lysosomal dysfunction, thus contributing to cell death, as observed in murine and rat β-cells and human pancreatic islets after chronic treatment with 400 µM palmitate and mouse astrocytes (inhibition of autophagic activity was observed after both acute and chronic palmitate treatment) [254,255]. The impairment of autophagy induced by chronic FFA treatment (500 µM palmitate) in mouse myoblasts may play a pivotal role in inflammation regulation and OS might also serve as a mechanism of autophagy modulation by FFA [256]. A study conducted by Mei et al. on human hepatocytes suggests a concept that saturated FFAs induce apoptosis, decrease autophagy, and that autophagy and apoptosis are two antagonistic events that tend to inhibit each other [257]. Even though the views on the ability of FFA to induce or inhibit autophagy are divided, there is no doubt that autophagy is an important aspect of the lipotoxicity of the cell. Because of its protective actions, any autophagy inhibiting factor will, without a doubt, lead to the worsening of lipotoxicity, therefore naming those factors as beneficial for understanding the lipotoxic impact on the cell.

## 4. Conclusions

Although saturated fatty acids play a vital role in organelle structure, signaling pathways, and supplying the energy demand, the excess over the cell’s ability to store fuel is proven to be detrimental for the proper functioning of the cell. Excess FFA might be caused by obesity, a high-fat diet, or dysfunction of adipose tissue manifested as reduced FFA storage capacity and increased releases of FFA from adipocytes. Lipotoxicity is referred to as deleterious processes caused by excess FFA in non-adipose tissues, such as OS, ER stress, inflammation, or insulin resistance. Despite many attempts to ascribe a specific metabolic pathway to the mechanism of toxic FFA actions, our review shows that lipids modulate cellular functions by multiple methods, resulting in many simultaneously occurring disturbances that tend to interfere with each other, creating a potential branched vicious cycle of self-driven lipotoxic processes. Therefore, being able to ameliorate lipotoxicity seems to be extremely difficult, because silencing one deleterious signaling cascade does not ensure that other signals will not disturb cell function. It seems that reducing excess FFA is the most eligible method to avoid toxic consequences.

Further understandings of the interrelation of molecular mechanisms of lipotoxicity may provide beneficial information about the development of pathological states. In addition to most obvious lipotoxicity-related diseases, such as T2DM or NAFLD, mechanisms of lipotoxicity may elucidate yet unknown etiologies of other diseases that are, at first glance, not related in any way to lipid metabolism, which constitutes further interesting research directions in this field.

## Figures and Tables

**Figure 1 cells-11-00844-f001:**
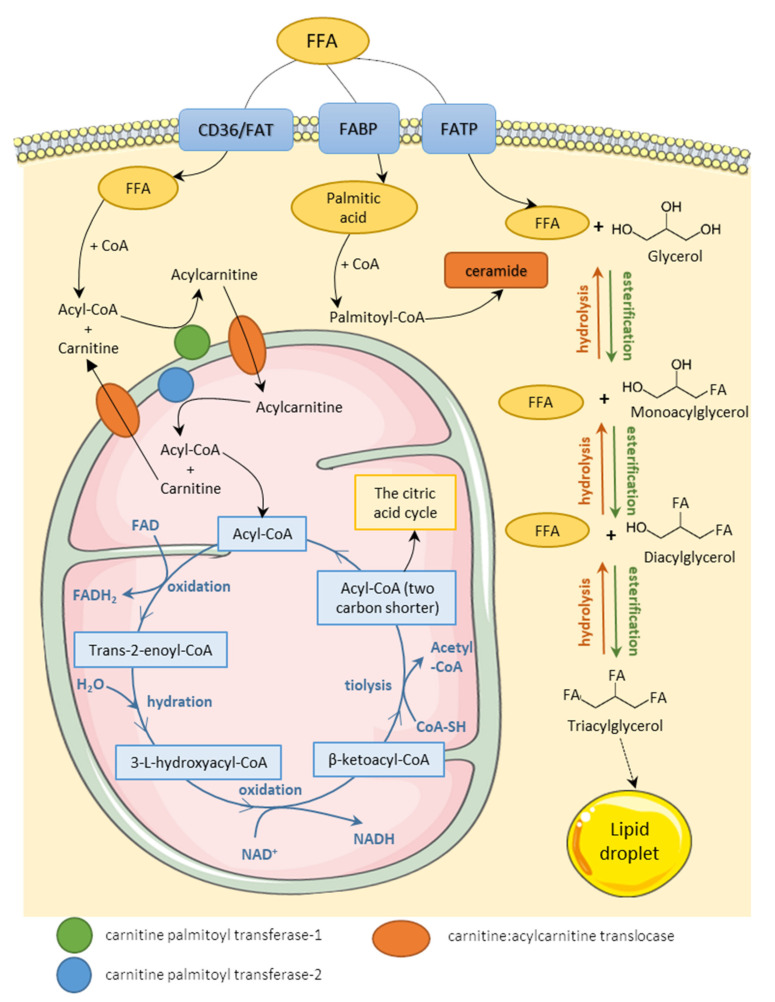
Homeostasis and metabolism of free fatty acids. FFAs circulating in the blood serum are bound to albumin. After the release from the albumin complex, FFAs are transported through the cell membrane by three various transporters: CD36/FAT, FABP, or FATP. Once in the cytoplasm, FFAs undergo esterification to TAG, which may be hydrolyzed to DAG. TAG is stored in lipid droplets. Lipid droplet is surrounded by proteins called perilipins, which regulate TAG exposition to hydrolytic enzymes. FFA may also be activated to acyl-CoA via the reaction with CoA. Palmitic acid is activated to palmitoyl-CoA, which is a substrate to de novo ceramide synthesis. Acyl-CoA is transported via the outer mitochondrial membrane by CPT-1, which also catalyzes acyl-CoA’s reaction with L-carnitine to create acyl-carnitine. The form of acyl-carnitine is necessary for the transport through the inner mitochondrial membrane via CACT. There, acyl-carnitine is disassembled via CPT-2 to acyl-CoA and L-carnitine. L-carnitine is transported back through the inner mitochondrial membrane via CACT. Acyl-CoA undergoes β-oxidation by releasing FADH2 and NADH, which serve as a substrate for ETC and acetyl-CoA, which is a substrate for the TCA cycle. Acetyl-CoA is also carboxylated to malonyl-CoA. FFA, free fatty acids. CD36/FAT, a cluster of differentiation 36/fatty acid transporter. FABP, fatty acid binding protein. FATP, fatty acid transport protein. TAG, triglyceride. DAG, diacylglycerol. Acyl-CoA, acyl-coenzyme A. CoA, coenzyme A. CPT-1, carnitine palmitoyl transferase-1. CACT, carnitine: acylcarnitine translocase. CPT-2, carnitine palmitoyl transferase-2. ETC, electron transport chain. Acetyl-CoA, acetyl coenzyme A. TCA cycle, the citric acid cycle [16,18,19,20].

**Table 1 cells-11-00844-t001:** Selected molecular mechanisms of lipotoxic oxidative stress and their sites of confirmed occurrence.

Molecular Mechanism of Oxidative Stress	Organism	Cell Culture	Organ/Cell Type	Reference
Increased general ROS production	human	isolated chondrocytes	cartilage	[80]
		cell line Hep2G	liver	[81]
	rat	isolated myocytes	muscle	[77]
		cell line H9c2	heart	[82]
		cell line INS-1	pancreas	[91]
	mouse	isolated cardiomyocytes	heart	[82]
		isolated podocytes	kidney	[89]
Increased mitochondrial ROS production	rat	cell line L6	muscle	[85]
		cell line H9c2	heart	[63]
	mouse	isolated cardiomyocytes	heart	[82]
		isolated podocytes	kidney	[92]
NOX activation	human	cell line Hep2G	liver	[81]
		isolated chondrocytes	cartilage	[80]
	rat	isolated myocytes	muscle	[77]
		cell line H9c2	heart	[83]
	mouse	isolated cardiomyocytes	heart	[82]
Reduction in ETC	rat	isolated myocytes	muscle	[77]
	mouse	isolated cardiomyocytes	heart	[82]
Reduction in MtMP *	human	HUVEC	endothelium	[84]
	rat	cell line H9c2	heart	[82]
	mouse	isolated podocytes	kidney	[92]
Reduced ATP generation	rat	cell line H9c2	heart	[63]
		cell line L6	muscle	[85]
	mouse	isolated podocytes	kidney	[92]
Iron-mediated toxicity	human	HUVEC	endothelium	[88]
increase in mitochondrial Ca^2+^	rat	cell line H9c2	heart	[82]

* MtMP—mitochondrial membrane potential.

**Table 2 cells-11-00844-t002:** Selected molecular mechanisms of lipotoxic ER stress and their sites of confirmed occurrence.

Molecular Mechanism of ER Stress	Organism	Cell Culture	Organ/Cell Type	Reference
Activation/phosphorylation of PERK	human	isolated hepatocytes	liver	[119]
		cell line L02	liver	[124]
		cell line Hep2G	liver	[124]
		cell line SH-SY5Y	neuroblastoma	[125]
	rat	isolated adipocytes	adipose tissue	[99]
		isolated hepatocytes	liver	[99]
		cell line INS-1	pancreas	[110,130]
		isolated neonatal rat cardiomyocytes	heart	[111]
	mouse	cell line RAW 264.7	macrophage	[116,117]
		cell line N2a	neuroblastoma	[125]
		cell line MIN-6	pancreas	[131,134]
Activation/phosphorylation of eIF2α	human	isolated hepatocytes	liver	[100]
		cell line Hep2G	liver	[146]
	rat	isolated adipocytes	adipose tissue	[99]
		isolated hepatocytes	liver	[99]
		cell line INS-1	pancreas	[130]
		cell line BRIN-BD11	pancreas	[118]
	mouse	cell line MIN-6	pancreas	[134]
XBP1 splicing/XBP1s increased expression or activation	human	cell line Hep2G	liver	[123,124]
		isolated β-cells	pancreas	[131]
		cell line SH-SY5Y	neuroblastoma	[125]
	rat	isolated adipocytes	adipose tissue	[99]
		isolated hepatocytes	liver	[99]
		cell line INS-1	pancreas	[110]
	mouse	isolated podocytes	kidney	[101]
		cell line AML12	liver	[113]
		cell line 3T3-L1	preadipocytes	[116]
		cell line RAW 264.7	macrophages	[116,117]
		cell line C2C12	muscle	[115]
		cell line N2a	neuroblastoma	[125]
		cell line MIN-6	β-cell	[131]
CHOP expression	human	isolated hepatocytes	liver	[100]
		cell line Hep2G	liver	[123,124]
		cell line L02	liver	[124]
		cell line SH-SY5Y	neuroblastoma	[125]
		isolated β-cells	pancreas	[131]
	rat	cell line INS-1	pancreas	[110,130]
		cell line BRIN-BD11	pancreas	[118]
		isolated cardiomyocytes	heart	[111]
	mouse	cell line 3T3-L1	preadipocytes	[116]
		cell line RAW 264.7	macrophages	[116,117]
		cell line C2C12	muscle	[115]
		cell line N2a	neuroblastoma	[125]
		cell line MIN-6	pancreas	[131,134]
Activation/phosphorylation of IRE1	human	cell line Hep2G	liver	[146]
		isolated hepatocytes	liver	[119]
		cell line SH-SY5Y	neuroblastoma	[125]
	rat	isolated cardiomyocytes	heart	[111]
		cell line INS-1	pancreas	[130]
	mouse	cell line AML12	liver	[113]
		cell line C2C12	muscle	[115]
		cell line N2a	neuroblastoma	[125]
Activation/phosphorylation of ATF4	human	cell line Hep2G	liver	[123,124]
		cell line L02	liver	[124]
		cell line SH-SY5Y	neuroblastoma	[125]
	rat	cell line INS-1	pancreas	[110]
		cell line BRIN-BD11	pancreas	[118]
	mouse	cell line C2C12	muscle	[115]
		cell line N2a	neuroblastoma	[125]
Activation/phosphorylation of ATF3	human	cell line SH-SY5Y	neuroblastoma	[125]
	rat	cell line INS-1	pancreas	[130]
	mouse	cell line N2a	neuroblastoma	[125]
mTORC1 activation	mouse	cell line AML12	liver	[113]
Perturbation of protein trafficking	mouse	cell line MIN-6	pancreas	[128,129]
ER Ca^2+^ depletion	human	isolated β-cells	pancreas	[131]
	rat	cell line INS-1	pancreas	[130]
		cell line H4IIEC3	liver	[133]
	mouse	cell line MIN-6	pancreas	[131,134]

**Table 3 cells-11-00844-t003:** Selected molecular mechanisms of lipotoxic inflammatory response and their sites of confirmed occurrence.

Molecular Mechanism of Inflammation	Organism	Cell Culture	Organ/Cell Type	Reference
Increased IL-6	human	HCASMC	smooth muscle	[147]
		hCAEC	endothelium	[148]
	rat	cell line L6	muscle	[149]
	mouse	cell line RAW 264.7	macrophage	[155]
		cell line BV-2	microglia	[178]
		cell line C2C12	muscle	[177]
Increased IL-1beta	human	HCASMC	smooth muscle	[147]
		cell line THP-1	monocyte	[168]
	mouse	cell line RAW 264.7	macrophage	[176]
		cell line BV-2	microglia	[178]
		BMDC	dendritic cells	[112]
		isolated Kupffer Cells	macrophages	[188]
Increased IL-8	human	HCASMC	smooth muscle	[147]
		hCAEC	endothelium	[148]
	mouse	cell line RAW 264.7	macrophage	[167]
Increased TNF-alfa	human	HCASMC	smooth muscle	[147]
		cell line Hep2G	liver	[195]
	mouse	cell line RAW 264.7	macrophage	[155]
		cell line 3T3-L1	preadipocyte	[156]
		cell line BV-2	microglia	[178]
		cell line C2C12	muscle	[191]
Activation of NF-kB	human	primary HAEC	endothelium	[150]
		cell line THP-1	monocyte	[168]
		cell line Hep2G	liver	[192]
	rat	cell line L6	muscle	[149]
		cell line INS-1	pancreas	[166]
	mouse	cell line C2C12	muscle	[177]
Activation of JNK	human	hCAEC	endothelium	[148]
		cell line THP-1	monocyte	[168]
	rat	cell line INS-1	pancreas	[166]
	mouse	cell line RAW 264.7	macrophage	[176]
		cell line C2C12	muscle	[177]
Involvement of TLRs	human	cell line THP-1	monocyte	[168]
	rat	cell line INS-1	pancreas	[166]
	mouse	cell line RAW 264.7	macrophage	[176]
		cell line BV-2	microglia	[178]
		cell line C2C12	muscle	[177]
Inflammasome activation	mouse	BMDC	dendritic cells	[112]
		cell line Hepa1–6	hepatoma	[186]
		cell line RAW 264.7	macrophage	[186]
		isolated Kupffer Cells	macrophage	[188]
Induced COX-2	mouse	cell line RAW 264.7	macrophage	[182]
Increased MCP-1	human	HCASMC	smooth muscle	[147]
		hCAEC	endothelium	[148]
	mouse	cell line RAW 264.7	macrophage	[167]

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
