# Peer review of "Molecular Mechanism of Lipotoxicity as an Interesting Aspect in the Development of Pathological States—Current View of Knowledge"

_cells, 2022, doi:10.3390/cells11050844_

Round 1
Reviewer 1 Report
Lipke et al. review molecular mechanisms of lipotoxicity aiming at a description of current knowledge in a cell-type-independent manner. This attempt to deduce emergent properties of lipotoxicity is an interesting concept.
After a general (i.e. historical) introduction, the authors elaborate on free fatty acid (FFA) metabolism under homeostatic conditions before they dive deep into the intracellular mechanisms of lipotoxicity.
Overall, I think the review provides a comprehensive overview of cellular mechanisms of lipotoxicity. I suggest to rework the manuscript strengthening that aspect even further by highlighting differences in lipotoxicity depending on tissue-specific niches. E.g. saying CD36 is a fatty acid translocase, with differential expression in liver as compared to the gastrointestinal tract.
In this regard, I think the manuscript is acceptable for publication after a minor revision. Please find my specific comments below:
In line 213 f., the authors write about signal transduction pathways and give "differentiation, proliferation, gene expression, membrane transport" as examples. However, these terms are rather generic and the outcome of intracellular signal transduction events. I therefore suggest to replace them with concrete examples of signal transduction pathways such as MAP kinase or PI3K/Akt.
In line 363, the authors are referring to IRE1 as inositol requiring 1. It should be inositol requiring enzyme 1.
In chapter 3.3, the authors need to explain better the interdependency between UPR and ER stress. First, they describe that ER stress causes UPR (line 361). Then, they say UPR causes ER stress (line 367).
In line 396, the authors mention mTOR as master regulator of cell growth, rightfully so. Subsequently, they describe that FFA lead through eIF2a-phosphorylation to the downregulation of translation. The authors should discuss, how this reduction in proteinbiosynthesis could counteract FFA-induced protein accumulation and ER stress (line 429). Furthermore, the authors could reflect on the discrimination of altered cellular protein content for cell growth and altered lipid content for membrane growth.
In chapter 3.4, the authors should explicitly differentiate between cell-intrinsic inflammation (i.e. cell-autonomous) or systemic inflammation, as they mention it correctly in line 492 f.
Chapter 3.6. on lipoapoptosis is mostly redundant. Any additional information could be fused to the respective sections on ER stress, OS, and mitochondrial metabolism.
In Figure 1, the authors write "The citric acid cycle", whereas in the text they are also referring to the Krebs cycle (line 142). They should be consistent in wording.
I don't find Figure 2 very informative. There are no details (e.g. signalling pathways) given and intracellular processes (e.g. ER stress) and systemic processes (e.g. inflammation, insulin resistance) are mixed up. One could argue about a direct connection between inflammation and insulin resistance. I would rather leave out that figure and include these terms in Figure 1 (e.g. pointing from DAG to ER stress...)
In general, I find that the authors use the expression "vicious circle/cycle" too often. They should stick to the concept of feedback loops but not suggest implicitly that two processes always automatically crank up one another, because as they otherwise show nicely, everything is connected, interdependent and therefore complicated.
Author Response
Dear Reviewer,
Underneath you can find enclosed explanation and description of action that has been undertaken in our article to convince you about the importance and valence of our work. We hope that after all the amendments that we have made you will find our paper interesting and valuable for publishing in Cells.
Point 1: “Overall, I think the review provides a comprehensive overview of cellular mechanisms of lipotoxicity. I suggest to rework the manuscript strengthening that aspect even further by highlighting differences in lipotoxicity depending on tissue-specific niches. E.g. saying CD36 is a fatty acid translocase, with differential expression in liver as compared to the gastrointestinal tract.”
Response 1: We agree about the importance of tissue specificity of discussed molecular mechanisms, therefore we tried to emphasize more that different types of cells are taken into account creating this review. We chose to present the data about certain mechanisms occurring in different tissues in the form of tables summarizing the most important lipotoxic molecular mechanisms. We believe that further extension of the aspects of tissue specificity might serve as a basis for new research or review, but unfortunately, we lucked the time to rework our manuscript more towards those aspects.
Point 2: “In line 213 f., the authors write about signal transduction pathways and give "differentiation, proliferation, gene expression, membrane transport" as examples. However, these terms are rather generic and the outcome of intracellular signal transduction events. I therefore suggest to replace them with concrete examples of signal transduction pathways such as MAP kinase or PI3K/Akt.”
Response 2: As suggested, the examples were changed to more specific examples.
Point 3: “In line 363, the authors are referring to IRE1 as inositol requiring 1. It should be inositol requiring enzyme 1.”
Response: The missing word “enzyme” was an oversight. We kindly thank you for pointing this out, so we were able to correct it.
Point 4: “In chapter 3.3, the authors need to explain better the interdependency between UPR and ER stress. First, they describe that ER stress causes UPR (line 361). Then, they say UPR causes ER stress (line 367).”
Response 4: After consideration, we found that this interdependency was indeed not very well explained. We made an effort to improve this chapter so that it would be more comprehensible, leaving no doubt about the relation between UPR and ER stress.
Point 5: “In line 396, the authors mention mTOR as master regulator of cell growth, rightfully so. Subsequently, they describe that FFA lead through eIF2a-phosphorylation to the downregulation of translation. The authors should discuss, how this reduction in proteinbiosynthesis could counteract FFA-induced protein accumulation and ER stress (line 429). Furthermore, the authors could reflect on the discrimination of altered cellular protein content for cell growth and altered lipid content for membrane growth.”
Response 5: Corrections were made and more attention has been paid to carefully and in detail explain the described molecular mechanisms. After consideration, there was no emphasis on the fact that the activation of mTORC1 by FFA in the context of lipotoxicity serves as a signaling pathway rather than leading to actual cell growth. Furthermore, in the case of ER stress, the phosphorylation state of mTORC1 is repressed by the activation of UPR, leading eventually to a decrease in protein synthesis. As for protein overload, this phenomenon was proposed as a mechanism of FFA-induced ER stress mainly in the case of β-cells. Furthermore, we believe that the inhibition of protein synthesis is an adaptive mechanism that is supposed to rescue ER stress, however during FFA abundance it is ineffective. We modified the chapter with the hope that there is an improvement in the explanation of those phenomenons.
Point 6: “In chapter 3.4, the authors should explicitly differentiate between cell-intrinsic inflammation (i.e. cell-autonomous) or systemic inflammation, as they mention it correctly in line 492 f.”
Response 6: As suggested, more emphasis has been put on the matter of which kind of inflammation is discussed.
Point 7: “Chapter 3.6. on lipoapoptosis is mostly redundant. Any additional information could be fused to the respective sections on ER stress, OS, and mitochondrial metabolism.”
Response 7: We agree with the reviewer. After consideration, chapter 3.6 was removed and a few pieces of information, that we find nonredundant, were placed in different chapters with the right context.
Point 8: “In Figure 1, the authors write "The citric acid cycle", whereas in the text they are also referring to the Krebs cycle (line 142). They should be consistent in wording.”
Response 8: As suggested, the name of the cycle was changed to “the citric acid cycle” in the whole manuscript.
Point 9: “I don't find Figure 2 very informative. There are no details (e.g. signalling pathways) given and intracellular processes (e.g. ER stress) and systemic processes (e.g. inflammation, insulin resistance) are mixed up. One could argue about a direct connection between inflammation and insulin resistance. I would rather leave out that figure and include these terms in Figure 1 (e.g. pointing from DAG to ER stress...)”
Response 9: After consideration, Figure 2 was removed from the manuscript. Indeed, it provides little specific information and we hope that the whole concept of the vicious cycle and the interplay between lipotoxic molecular mechanisms were better explained in the text.
Point 10: “In general, I find that the authors use the expression "vicious circle/cycle" too often. They should stick to the concept of feedback loops but not suggest implicitly that two processes always automatically crank up one another, because as they otherwise show nicely, everything is connected, interdependent and therefore complicated."
Response 10: The use of the expression “vicious circle/cycle” was reduced, as suggested. After consideration, we found that this expression was indeed overused. Thank you for this comment.
We kindly thank you for your comments which we found very helpful. We hope that the introduced changes will improve the quality of our manuscript and allow it to be published in the Cells journal.
Best regards,
Katarzyna Lipke
Reviewer 2 Report
The review manuscript by Lipke at. al. describes the pathological effects of an excess of free fatty acids, mainly palmitic acid, on cells. The authors announce in the title that they will discuss the molecular mechanism of lipotoxicity, focusing on a comprehensive description of the cellular mechanism. However, focusing on a comprehensive mechanism is incorrect because each cell type/tissue has a specific property that must be considered. Cells/tissues differ in their ability to deal with stress, either oxidative or ER stress, and respond accordingly to FFA excess and signaling. So it is well known that FFA overload leads to production of ROS, induction of ER stress and inflammation, but, it is very important to mention this in terms of individual cell types/tissues. This would make the review more valuable to the scientific community. Also, it does not bring any new findings and is more of a general book chapter text.
Although FAO is very important in FFA signaling, I miss information about FFA receptor (FFAR)-mediated signaling molecules such as GPRs that play an important signaling role. FFA is not just an energy substrate.
FFA are processed to LC acyl-coA, which plays an important regulatory role in cells and is further represented by protein S-acylation. This affects many ion channels and transporters in cells. This needs to be reviewed in the manuscript. The involvement of different ion channels in FFA signaling has far-reaching consequences for metabolic activity as well as the polarization of cell membranes, exocytosis etc.
Figure. 2 represents very general and widely accepted knowledge. I would therefore like to see the review focus more on specifics of individual tissues and this summarize in a table or a figure.
In general, it is important to be aware that the experimental amounts of FFA supplied to cells or animals represent an excess that is quite different from physiological or chronic pathological conditions. Therefore, experiments should be interpreted with caution.
Author Response
Dear Reviewer,
Underneath you can find enclosed explanation and description of action that has been undertaken in our article to convince you about the importance and valence of our work. We hope that after all the amendments that we have made you will find our paper interesting and valuable for publishing in Cells.
Point 1: “The review manuscript by Lipke at. al. describes the pathological effects of an excess of free fatty acids, mainly palmitic acid, on cells. The authors announce in the title that they will discuss the molecular mechanism of lipotoxicity, focusing on a comprehensive description of the cellular mechanism. However, focusing on a comprehensive mechanism is incorrect because each cell type/tissue has a specific property that must be considered. Cells/tissues differ in their ability to deal with stress, either oxidative or ER stress, and respond accordingly to FFA excess and signaling. So it is well known that FFA overload leads to production of ROS, induction of ER stress and inflammation, but, it is very important to mention this in terms of individual cell types/tissues. This would make the review more valuable to the scientific community. Also, it does not bring any new findings and is more of a general book chapter text.”
Response 1: We agree with the reviewer, that putting emphasis on tissue specificity is essential. We made effort to underline this specificity both in the text and in the form of tables presenting which molecular mechanisms occur in which kind of cells/tissues with a comment. We believe that further extension of the aspects of tissue specificity might serve as a basis for new research or review. We hope that Reviewers will find our improvements sufficient. Unfortunately, we lucked time to rework the whole text taking into account the breadth of the topic.
Point 2: “Although FAO is very important in FFA signaling, I miss information about FFA receptor (FFAR)-mediated signaling molecules such as GPRs that play an important signaling role. FFA is not just an energy substrate.”
Response 2: As suggested, there was an additional review made about the FFA receptor in the context of lipotoxicity. A new paragraph was created dedicated exclusively to the FFA receptor aspect.
Point 3: “FFA are processed to LC acyl-coA, which plays an important regulatory role in cells and is further represented by protein S-acylation. This affects many ion channels and transporters in cells. This needs to be reviewed in the manuscript. The involvement of different ion channels in FFA signaling has far-reaching consequences for metabolic activity as well as the polarization of cell membranes, exocytosis etc.”
Response 3: After consideration, we found that the role of protein S-acylation was indeed missing from our manuscript as it is an important aspect of FFA role in the cell and lipotoxicity. Thank you for this comment. Further review was made about the role of protein S-acylation and a new paragraph was added to the manuscript.
Point 4: “Figure. 2 represents very general and widely accepted knowledge. I would therefore like to see the review focus more on specifics of individual tissues and this summarize in a table or a figure.”
Response 4: After consideration, Figure 2 was removed from the manuscript. Indeed, it provides little specific information. Three new tables were added to the text instead of Figure 2.
We kindly thank you for your comments which we found very helpful. We hope that the introduced changes will improve the quality of our manuscript and allow it to be published in the Cells journal.
Best regards,
Katarzyna Lipke
Round 2
Reviewer 2 Report
The revision has greatly improved the manuscript. It is helpful that the authors have included tables of given mechanisms in selected tissues. However, they should go through the entire text to include tissues and animal species in the description of a given effect so that the reader does not have to look in the references to see which tissue/species the mechanism described is valid for. This applies, for example, to line 439 (ref 114), line 736 (ref 82,227), line 740 (ref229), etc.
I would also appreciate if the authors could define the effect of FFA in terms of duration (acute/chronic) and if possible concentration. This is very important in terms of the effect on cells/tissues and is not in the text.
The chapter on oxidative stress is rather weak and does not include information on the mechanism of mitochondrial dysregulation, e.g. uncoupling or UCP activity.
There are some typos that need to be corrected.
Author Response
Response to Reviewer 2 (Round 2) Comments
Point 1: The revision has greatly improved the manuscript. It is helpful that the authors have included tables of given mechanisms in selected tissues. However, they should go through the entire text to include tissues and animal species in the description of a given effect so that the reader does not have to look in the references to see which tissue/species the mechanism described is valid for. This applies, for example, to line 439 (ref 114), line 736 (ref 82,227), line 740 (ref229), etc.
Response 1: As suggested, we made further effort to underline types of tissues and animal species in the description of the given effect.
Point 2: I would also appreciate if the authors could define the effect of FFA in terms of duration (acute/chronic) and if possible concentration. This is very important in terms of the effect on cells/tissues and is not in the text.
Response 2: As suggested, we went through the entire text and included terms of the duration of the FFA effect. FFA concentrations were also added.
Point 3: The chapter on oxidative stress is rather weak and does not include information on the mechanism of mitochondrial dysregulation, e.g. uncoupling or UCP activity.
Response 3: After consideration, we agree with the reviewer. We chose to remodel the text by removing the paragraph about mitochondrial dysfunction from chapter 3.1 and placing it in chapter 3.2 „Oxidative stress”. Furthermore, as suggested, the process of mitochondrial dysfunctions, uncoupling, and UCP activity was further explained.
Point 4: There are some typos that need to be corrected.
Response 4: We have put additional care to find and fix typos in the text. Also, an additional grammar check was performed.
We hope that reviewers will find our revised manuscript suitable for publication in Cells.